# Recognition of intracranial hypertension using handheld optical coherence tomography in children (RIO Study): a diagnostic accuracy study protocol

Sohaib R Rufai [1,2,3] Noor ul Owase Jeelani,[3,4] Richard Bowman [1,3]
Catey Bunce [5] Frank A Proudlock,[2] Irene Gottlob[2]

¹Clinical and Academic Department of Ophthalmology, Great Ormond Street Hospital For Children NHS Foundation Trust, London, UK
²Ulverscroft Eye Unit, University of Leicester, Leicester, UK
³UCL Great Ormond Street Institute of Child Health, London, UK
⁴Craniofacial Unit, Great Ormond Street Hospital for Children NHS Foundation Trust, London, UK
⁵Clinical Trials Unit, Royal Marsden Hospital NHS Trust, London, UK

**Correspondence to**
Dr Sohaib R Rufai;
Sohaib.Rufai@nhs.net

## ABSTRACT

**Introduction** Paediatric intracranial hypertension (IH) is a rare but serious condition that can pose deleterious effects on the brain and vision. Estimating intracranial pressure (ICP) in children is difficult. Gold standard direct ICP measurement is invasive and carries risk. It is impractical to routinely perform direct ICP measurements over time for all children at risk of IH. This study proposes to assess the diagnostic accuracy of handheld optical coherence tomography (OCT), a non-invasive ocular imaging method, to detect IH in children.

**Methods and analysis** This is a prospective study evaluating the diagnostic accuracy of handheld OCT for IH in at risk children. Inclusion criteria include clinical and/or genetic diagnosis of craniosynostosis, idiopathic intracranial hypertension, space occupying lesion or other conditions association with IH and age 0–18 years old. Exclusion criteria include patients older than 18 years of age and/or absence of condition placing the child at risk of IH. The primary outcome measures are handheld OCT and 48-hour ICP assessments, which will be used for diagnostic accuracy testing (sensitivity, specificity, positive predictive value, negative predictive value and accuracy). Main secondary outcome measures include visual acuity, fundoscopic examination, contrast sensitivity, visual field testing and visual evoked potentials, wherever possible.

**Ethics and dissemination** Ethical approval was granted for this study by the East Midlands Nottingham 2 Research Ethics committee (UOL0348/IRAS 105137). Our findings will be disseminated through presentation at relevant meetings, peer-reviewed publication and via the popular media.

**Trial registration number** ISRCTN52858719

## INTRODUCTION
### Intracranial hypertension
First described by Quincke in 1896, intracranial hypertension (IH) remains an area of major importance among the medical profession.[1] If undetected and untreated, IH can pose devastating consequences including visual impairment, cognitive impairment, disability and even death.[2][3] This is of particular significance in subacute conditions affecting children, because insidious elevation of intracranial pressure

### Strengths and limitations of this study

► This the first diagnostic accuracy study protocol using handheld optical coherence tomography (OCT) to detect intracranial hypertension in at risk children.
► This protocol promotes transparency of our methodology and reduces the risk of duplication.
► Fully automated handheld OCT image analysis is not currently available and manual segmentation takes longer, although our evidence-based methodology features semiautomated segmentation.
► Although handheld OCT can provide in vivo imaging in infants without sedation, it could be especially challenging in this patient population, which could limit imaging success rates.
► Gold standard intracranial pressure measurements form our reference standard, but these are only performed in select patients, thus limiting our overall recruitment.

(ICP) may only be detected after insult to the brain and vision has already begun.[2][4] Therefore, it is crucial that IH is detected and treated as early as possible to prevent the sequelae of IH.

The incidence of IH is approximately 0.63–0.71 per 100 000 children.[5][6] IH can be categorised as primary, also known as idiopathic intracranial hypertension (IIH), and secondary; the former is characterised by elevated ICP with no identifiable cause, while the latter is characterised by elevated ICP directly resulting from another condition.[4][7] Common associations of primary IH include female sex, high body mass index (BMI) and postpubertal status; however, it can occur in children with no predilection for sex or high BMI.[4][8–14] Examples of secondary IH include expansive skull pathology, that is, space occupying lesion or hydrocephalus, or constricting skull pathology, such as craniosynostosis, characterised by the premature fusion of cranial sutures.[2–4]

## Measuring ICP

Measuring ICP in children presents a number of challenges. The gold standard method is direct intraparenchymal measurement, but this involves hospital admission and carries significant risks including infection, bleeding, cerebrospinal fluid (CSF) leak, mechanical failure and general anaesthesia risks.[15] Another limitation of intraparenchymal measurement is the difficulty in obtaining repeated measures over time. Furthermore, there is no clinical consensus on timing, frequency and duration for accurate measurement, or what figure constitutes raised ICP in children. Many consider ICP greater than 15 mm Hg to be elevated,[2 16–24] but Hayward et al[25] argue this could reasonably be raised to 20 mm Hg.[25] Overnight ICP monitoring during natural sleep, which provides a 'grand average' with description of the dynamics of the ICP, is considered the gold standard.[26] In Great Ormond Street Hospital for Children (GOSH), London, monitoring is performed using an ICP bolt (Raumedic AG, Helmbrechts, Germany) for 48 hours according to clinical discretion, particularly in children with clinical suspicion of IH where ophthalmological findings, including fundoscopic and electrodiagnostic findings, are equivocal. This represents an important patient cohort where a more sensitive non-invasive measure could improve clinical decision-making and reduce the need for invasive ICP bolt monitoring. However, the youngest in this group undergoing ICP bolt monitoring are typically toddlers and older, whereas those aged under 1 year with conditions such as craniosynostosis typically undergo surgical treatment expectantly rather than having ICP bolt monitoring.

The ideal measure of ICP would be non-invasive, acceptable for children, quantitative and would permit serial measurements to appreciate evolving changes related to rising ICP. Existing surrogate measures of ICP do not fulfil all these criteria, including fundoscopy,[27] radiography[28] and ocular ultrasonography,[29] each failing to provide adequate sensitivity to be used as screening tools. In particular, funduscopy is routinely used in paediatric ophthalmology clinics to detect IH. Tuite et al[27] demonstrated that fundoscopy yielded sensitivity of only 22% for IH in children with craniosynostosis aged under 8 years old, while specificity was 98%. Hence, papilloedema detected on fundoscopy reliably indicates IH in younger children with craniosynostosis, but crucially its absence does not exclude IH.

## Optical coherence tomography

The optic nerve is primarily intracranial; hence, IH can cause optic nerve swelling with secondary retinal changes. Optical coherence tomography (OCT) is a non-invasive imaging modality that can rapidly produce ultra-high-resolution cross-sectional images of the optic nerve and retina in vivo.[30] OCT scans of the optic nerve head (ONH) can provide highly reproducible quantitative measures associated with IH within seconds.[31] Adult studies have demonstrated the value of OCT in IH, including volumetric analysis.[32–34]

Our recent systematic review[35] revealed 21 studies using OCT to assist diagnosis and management of IH in children with craniosynostosis, IIH, space occupying lesion and other pathology. This review revealed generally good diagnostic capability for OCT, but no level 1 evidence was identified. There was inconsistency in OCT parameters and reference standards used, with no study featuring overnight ICP bolt monitoring. Validating prospective research is therefore required to determine optimal OCT parameters in this role and to develop formal clinical guidelines. Moreover, no studies used handheld OCT in children with craniosynostosis, IIH or space occupying lesion.

## Handheld OCT

OCT has revolutionised the clinical practice of ophthalmology over the past two decades. However, infants and young children have previously been largely deprived of this technology as conventional OCT machines were designed for adults, requiring compliance with a chin rest, good fixation and cooperation. These limitations have been overcome with the recent advent of the handheld spectral-domain OCT designed for infants, which is demonstrably reliable in awake infants.[30 36] Normal development of the optic nerve[37] and retina[38] in children has recently been described using handheld OCT. A wide range of conditions have also been investigated using handheld OCT in children, including retinopathy of prematurity,[39] nystagmus,[40] foveal hypoplasia,[41] optic nerve hypoplasia,[41] achromatopsia,[42] albinism,[43] primary congenital glaucoma,[44] microcephaly,[45] craniopagus twins[46] and others.

More recently, we performed a feasibility and repeatability study[47] using handheld OCT in children with craniosynostosis (n=50, median age 51 months, age range 2–157 months). We found good feasibility, with 86% achieving at least unilateral OCT and 76% achieving bilateral OCTs of the ONH. This was higher than the success rate in healthy children found by Patel et al[37] (70% achieving at least unilateral OCT imaging of the ONH). Factors boosting the likelihood of success in children with craniosynostosis included good understanding and cooperation of the child and parent/guardian and availability of an assistant. We also performed repeatability analysis in 20 children and found good repeatability (intraclass correlation coefficient range, 0.77–0.99; the majority exceeded 0.90).

Advantages of using handheld OCT in children at risk of IH include the following:
- Handheld OCT can be performed in awake babies and infants, without the need for general anaesthesia.
- Even if under general anaesthesia, handheld OCT can be performed in a child lying down, both in the supine and lateral decubitus positions.
- Handheld OCT scans can be easily repeated in the clinical setting, enabling the assessment of evolving changes over time.

► Handheld OCT semiautomated segmentation parameters can be fully controlled by the investigator.[37 38 47]

## OBJECTIVES
### Primary objective
To assess the diagnostic accuracy of OCT for paediatric IH.

### Secondary objectives
► To assess success rates of handheld OCT in children with craniosynostosis and other children at risk of IH.
► To evaluate ophthalmological outcomes, including visual acuity (VA), in this cohort and to compare these with corresponding OCT data.
► To characterise ONH and retinal morphology in conditions associated with paediatric IH.
► To compare repeated OCT data between clinic visits±ICP assessments±cranial vault expansion surgery±follow up, with ICP assessments wherever possible.
► To compare OCT data in children at risk of IH with that of healthy age-matched controls from our ongoing paediatric handheld OCT normative database.[37 38]

## METHODS
### Study design
Single-centre, prospective diagnostic accuracy study.

### Subjects
Children aged 0–18 who are diagnosed with conditions associated with IH, including craniosynostosis of all forms (syndromic and non-syndromic), IIH, space occupying lesion and others.

### Recruitment
Subjects will be recruited from the ophthalmology clinic at GOSH, London. In addition, subjects will be recruited from the admissions ward prior to 48-hour ICP assessment and/or cranial vault expansion surgery, should this be their first stage of clinical care when referred to GOSH.

### Inclusion criteria
► Patients aged 0–18.
► Clinical and/or genetic diagnosis of craniosynostosis, IIH, space occupying lesion or other conditions association with IH.

### Exclusion criteria
► Patients older than 18 years of age.
► Patients not wishing to participate.
► Patients incapable of giving consent and without a legal guardian willing or able to do so.

### Information and consent process
Patients presenting to their pre-existing clinical appointments at GOSH, along with their family members/guardians, will be approached by the study investigator(s) and will be provided with the study information leaflet. Information leaflets suitable for children are also available. If children over 7 years old are able to understand the study, they will be given the option to provide their signature as a form of assent on the consent form along with their parent/guardian's signature.

After the consent form is signed, patients will be taken to the investigator's clinic room, or the handheld OCT scanner will be brought to them if appropriate, in order to acquire their OCT images. If it is not possible to take their scan on the same day, patients will be invited back to have their OCT scan at a suitable date in the near future. Where possible, subjects will have repeat OCT scans in future appointments to obtain longitudinal data. Consent will be taken in advance of ICP assessment or any surgical procedure requiring general anaesthesia.

### Outcome measures
Our primary outcome measures are OCT examinations and ICP assessments. Our primary OCT outcome measure is retinal nerve fibre layer (RNFL) thickness; other OCT parameters are listed below. OCT examination must be performed at the time of ICP assessment, or within 28 days prior to or 14 days following ICP assessment. Only one visit per child for OCT examination is required to fulfil the primary objective.

Our secondary outcome measures include the following, where possible:
► Success rates of handheld OCT in children at risk of IH, particularly in craniosynostosis.
► Repeated OCT data between clinic visits±ICP assessments±cranial vault expansion surgery±follow up, with repeated ICP data if available.
► Fundoscopic examination within 3 months of OCT examination and/or at most recent follow-up visit.
► VA measured within 3 months of OCT examination and/or at most recent follow-up visit.
► Contrast sensitivity measured within 3 months of OCT examination and/or at most recent follow-up visit.
► Visual field testing within 3 months of OCT examination and/or at most recent follow-up visit.
► Visual evoked potentials (VEPs) within 3 months of OCT examination and/or at most recent follow-up visit.
► Comparison of the above measures with ICP and final VA, where available and appropriate.

### Handheld OCT image acquisition and analysis
A non-contact, spectral domain handheld OCT scanner (Envisu C-2300, 3.3 μm axial resolution; Leica Microsystems, Wetzlar, Germany) will be used to scan the ONH and fovea. A 12×8 mm scanning window will be used in the acquisition protocol—this permits imaging of both the ONH and fovea in one scan. The three-dimensional raster scan for both scan sequences will consist of 100 B-scans and 500 A-scans per B-scan line resulting in a short acquisition time (1.9 s) enabling imaging of the ONH and fovea with minimal movement artefact. Handheld OCT will be performed at the time of 48-hour ICP monitoring to fulfil our primary objective; wherever

possible, handheld OCT will also be performed during clinic visits to fulfil our secondary objectives. In awake children, visual devices including cartoons on smartphones and tablets will be used as appropriate. Where possible, an assistant can operate the controls to acquire images, otherwise the lead investigator (SRR) will acquire images alone using the foot pedal.

Our imaging protocol involves scanning the right eye followed by the left eye. Once the image is acquired, the en face view is used to identify the ONH, which is navigated frame by frame until the central slice featuring the deepest optic cup is identified for ONH analysis. For foveal segmentation, the fovea is navigated frame by frame until the central slice featuring the deepest foveal pit is identified. Scans are repeated twice per eye where possible, and any scans of non-analysable quality are discarded. Analysable quality is defined as an ONH tomogram wherein the edges of the disc margins and the cup profile, including its lowermost point, can be clearly visualised, or a foveal tomogram wherein it is possible to identify the foveal pit in full and all foveal layers from the external limiting membrane to Bruch's membrane. In cases of papilloedema, the handheld OCT probe can be repositioned slightly forwards in the axial plane, or the reference arm can be lowered, until the entire swollen ONH is visualised without missing the disc margins. Using this technique, all cases with papilloedema were successfully imaged in our recent feasibility and repeatability study.[47]

ImageJ V.1.48 (National Institutes of Health, Bethesda, Maryland)[48] shall be used for quantitative segmentation and the ABSnake plugin[49] shall be used identify the internal limiting membrane contour, which can be corrected manually where required. Lateral distance measurements (defined for adults on the machine) shall be corrected to account for the smaller axial lengths in children using a conversion table according to age from the data presented by Maldonado et al.[50] Optic nerve and retinal analyses will be based on single B-scans through the deepest part of the optic cup and foveal pit, respectively.

Cup and disc parameters will include the following:
► Cup depth.
► Cup width.
► Cup area.
► Disc width.

Lateral measurements will be expressed as visual angles rather than distances since disc and cup diameters remain relatively constant in the developing eye or with changing axial length.[37]

Rim parameters will include the following:
► Rim width.
► Rim height (nasal/temporal).
► Rim area.
► Nasal and temporal ppRNFL thickness.
► Total retinal thickness.
► Bruch's membrane opening-minimum rim width (nasal/temporal).
► Bruch's membrane orientation.

► Full peripapillary volumetric analysis can be performed wherever possible, using a recently published protocol.[51]

Where available, retinal parameters will include the following:
► Macular and perimacular retinal thickness.
► Foveal pit width.
► Foveal pit depth.
► Foveal pit area.
► Segmentation of all retinal layers.

## ICP assessment

As per standard clinical care at GOSH, patients are admitted for gold-standard, 48-hour ICP assessments. An intraparenchymal catheter and bolt system is used (RAUMEDIC Neurovent-P, RAUMEDIC AG, Helmbrechts, Germany). The neurosurgeon measuring ICP will be masked to the OCT findings. Due to lack of agreement over what ICP values constitute IH in children, two definitions will be used for IH:

Definition 1: Grand mean ICP $\geq$15 mmHg, or ICP traces with $\geq$2 ICP spikes at least double the baseline ICP and $\geq$15 mm Hg for $\geq$5 min.

Definition 2: Grand mean ICP $\geq$20 mmHg, or ICP traces with $\geq$2 ICP spikes at least double the baseline ICP and $\geq$20 mm Hg for $\geq$5 min.

ICP assessments using both definitions will be included in multilevel statistical analyses.

## Secondary outcome data assessment

Fundoscopic examination of the optic discs will be performed by experienced paediatric ophthalmologists and will be graded as follows: normal, papilloedema, mild pallor, severe pallor. Chart logMAR VA will be measured wherever possible using the Thomson Test Chart (Thomson Software Solutions, Hatfield, UK), which uses crowded letter optotypes. If the child is unable to take part in the Thomson Test Chart, then the Keeler logMAR picture test (Keeler, Windsor, UK) will be used . This test requires the child to match easily recognisable pictures (eg, house, car, apple) and is indicated in verbal children who cannot match letter optotypes.[52] In preverbal children, we will use Keeler Acuity Cards (Keeler) for preferential looking testing. Contrast sensitivity will be measured where possible using the Thomson Test Chart contrast sensitivity module (Thomson Software Solutions, Hatfield, UK), similar to the Pelli-Robson test. Visual field testing will be performed where possible using the Octopus 900 (Haag Streit UK Ltd, Harlow, UK) and/or Humphrey Field Analyzer 3 (Carl Zeiss Ltd, Cambridge, UK). Wherever possible, VEP assessments will be performed and graded according to a modified criterion created by Thompson et al.[53] In children with craniosynostosis monitored at GOSH, clinical assessments and VEPs are currently performed according to an ophthalmological surveillance protocol.[54]

## Sample size calculation

A cohort of at least 39 children is required to demonstrate at least 40% improvement in sensitivity compared with 22% by fundoscopy demonstrated by Tuite et al[27] (McNemar's Z-test: n=39; power=80%, α=0.05). This would demonstrate sensitivity of at least 62%, a conservative estimate comparable to the least sensitive OCT measure demonstrated by Swanson et al[16] (maximal retinal thickness, 63% sensitivity) for detecting IH in children with craniosynostosis.

## Statistical analysis

Diagnostic accuracy testing will be performed to calculate sensitivity, specificity, positive predictive value, negative predictive value and accuracy of OCT parameters to detect IH, using the OCT parameters and two definitions of IH described above. Receiver operating characteristic curves will be plotted to evaluate diagnostic accuracy of OCT parameters for IH. Mixed model regression analysis will be performed to assess the strength of association between OCT parameters and secondary outcome measures, described above. The eye recorded (right/left) will be included in the model, to address intereye correlations.

## Timing

Our study commenced in February 2020 but was halted in between March 2020 and September 2020 due to COVID-19 restrictions in the UK. Our study restarted in September 2020. We aim to finalise this study by February 2024.

## Assessment of possible adverse effects

OCT is a safe and non-contact imaging technique that uses low-coherence light. It is extremely unlikely that any adverse events could arise during OCT imaging. Standard clinical care will continue during and beyond the completion of the RIO study.

## Patient and public involvement statement

Patients, families and the public will be actively involved in the RIO study via two advisory groups: (1) the Young Person's Advisory Group (YPAG), including participants from the GOSH YPAG and Moorfields Eye-YPAG; (2) the GOSH Parent and Carer Advisory Group. Participants in both advisory groups will assist the RIO investigators in several ways, including but not limited to: (1) sharing their lived experience of the conditions being studied; (2) keeping the study child-friendly and parent-friendly; (3) keeping recruitment rates high; (4) dissemination of study findings. Both advisory groups will meet at least annually.

## Ethics and dissemination

Ethical approval was granted for this study by the East Midlands Nottingham 2 Research Ethics committee (UOL0348/IRAS 105137). Participants will already be under the care of GOSH, with continuity of care beyond our study. Our research clinic will involve rapid,

non-invasive OCT imaging during the patients' existing appointments + additional appointments only where required. Our study will not affect the overall management of the children. Consent will be sought from parents/guardians. Our findings will be disseminated via relevant local and national/international meetings, as well as via peer reviewed journal publication and social media.

**Acknowledgements** We would like to thank the participants of GOSH Young Persons Advisory Group (YPAG), Moorfields Eye YPAG and GOSH Parent and Carer Advisory Group for supporting this study.

**Contributors** SRR: concept, protocol writing, statistical methods, final approval. NuOJ and RB: concept, protocol writing, critical revision, final approval. CB, FAP and IG: concept, protocol writing, statistical methods, critical revision, final approval.

**Funding** This work is supported by the Medical Research Council, UK (Grant No. MR/N004566/1). Dr Sohaib Rufai is funded by the National Institute of Health Research Doctoral Fellowship (Award ID: NIHR300155). This work is supported by the NIHR Great Ormond Street Hospital Biomedical Research Centre. This protocol proposes independent research funded by the National Institute for Health Research (NIHR) and the Medical Research Council, UK.

**Disclaimer** The views expressed are those of the author(s) and not necessarily those of the MRC, the NHS, the NIHR or the Department of Health and Social Care.

**Competing interests** Dr. Proudlock reports personal fees from Leica Microsystems to run a virtual clinical symposium for hand-held optical coherence tomography, outside the submitted work. Dr. Rufai, Dr. Jeelani, Dr. Bowman, Dr. Bunce and Prof. Gottlob have no competing interests to report.

**Patient consent for publication** Not applicable.

**Provenance and peer review** Not commissioned; externally peer reviewed.

**ORCID iDs**
Sohaib R Rufai http://orcid.org/0000-0001-8134-6393
Richard Bowman http://orcid.org/0000-0001-5422-4104
Catey Bunce http://orcid.org/0000-0002-0935-3713

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
