## [Reviewer comments · BMJ Open]

ARTICLE DETAILS

TITLE (PROVISIONAL)	Recognition of Intracranial Hypertension using Handheld Optical Coherence Tomography in Children (The RIO Study): A Diagnostic Accuracy Study Protocol
AUTHORS	Rufai, Sohaib; Jeelani, Noor ul Owase; Bowman, Richard; Bunce, Catey; Proudlock, Frank; Gottlob, Irene

VERSION 1 – REVIEW

REVIEWER	Zimmermann, Hanna Charité Universitätsmedizin Berlin
REVIEW RETURNED	29-Mar-2021

GENERAL COMMENTS	This is an interesting and highly relevant study protocol. However, I think some processes should be described more clearly. Also I think the authors could still optimize their analysis strategy. - The definition of outcomes is not very clear. „The primary outcome measures are handheld OCT and 48-hour ICP assessments,...“ and later the authors list almost 20 different OCT parameters, However, „Primary outcome“ is supposed to be exactly 1 measure, so the authors might consider to define a specific OCT parameter or correlation as primary outcome.- The use of the \pm symbol is not very clear to me. E.g. „HH-OCT will be performed during initial and all follow-up appointments until 48-hour ICP monitoring + surgery + follow-up appointments where applicable.“ Please provide a very clear description of the baseline and follow up times, preferably as a chart. I think it is important that the time points are very standardized in order to allow for comparability.- The authors describe they are using a 12x8 mm scanning field for OCT. This is quite large, is this supposed to cover the macula and ONH in one scan, or are both regions scanned with this protocol individually?- Regarding OCT image analysis, I understand that the device they are using delivers no parameters and all analysis is done manually in ImageJ. This should be described in more detail.- While I agree with the authors that results on OCT in children with IP are sparse, there is a plethora of studies investigating OCT in adult patients. I believe it makes sense to refer more to those. Especially, the 3D analysis of ONH volume has been of particular diagnostic value:- https://www.ncbi.nlm.nih.gov/pmc/articles/PMC4266084/- Wang J-K, Kardon RH, Kupersmith MJ, Garvin MK. Automated quantification of volumetric optic disc swelling in papilledema using spectral-domain optical coherence tomography. Invest Ophthalmol Vis Sci. 2012; 53: 4069–4075.
---

	 - Albrecht P, Blasberg C, Ringelstein M, et al. Optical coherence tomography for the diagnosis and monitoring of idiopathic intracranial hypertension. J Neurol. 2017; 264: 1370–1380. - Kaufhold F, Kadas EM, Schmidt C, et al. Optic nerve head quantification in idiopathic intracranial hypertension by spectral domain OCT. PLoS One. 2012; 7: e36965. - https://tvst.arvojournals.org/article.aspx?articleid=2761893 While I understand that the HH-OCT device that is used does not provide volumetric analysis, the authors might consider to implement a method for that, in order to make maximum use of the data assessed.  - As visual fields demand a high degree of compliance and patient cooperation: Have the authors considered to use multifocal VEP as an objective measure of perimetry? - Acquisition of high-quality OCT data in the targeted patient cohort will be challenging because of the axial expansion of papilledema, and the limited compliance of children. How are the authors going to address this? - How are the authors going to address the growing eyeball size in children? - Statistical analysis: Can the authors comment how they are going to address intra-subject inter-eye correlations? - General: Please make sure to introduce acronyms when they are first mentioned, e.g. „IIH“ in abstract
--	---

REVIEWER	Swanson, Jordan The Children's Hospital of Philadelphia, Division of Plastic and Reconstructive Surgery
REVIEW RETURNED	13-Jul-2021

GENERAL COMMENTS	The authors seek to assess the use of handheld optical coherence tomography (OCT) to detect elevated intracranial pressure (ICP) in at-risk children, by examining retinal parameters in children who are undergoing direct ICP measurement. The study is well-conceived, and builds on evolving evidence which the authors present articulately. It has the potential to improve both clinical pathways and treatment decision-making in unique clinical cases, by improving the fidelity of non-invasive methods to evaluate ICP. Several additional considerations may strengthen the manuscript. Questions:  1. Pediatric ophthalmology centers (including ours) report difficulty and inconsistency in obtaining OCT data from awake children under 3 years of age. Although the authors cite two references from their team (30,35) of “reliably” using OCT in infants, it is not clear from either of these references the “success rate” of success compared to attempts at OCT acquisition in infants. This information, and/or practical guidance for how this can be feasibly undertaken, would strengthen the methods section. Does the number of prospective subjects to include need to be increased to account for this? 2. Direct ICP assessment in patients with craniosynostosis at many craniofacial centers is obtained selectively for atypical patients and those with suspected recurrence of cranio-cerebral disproportion. Both of these groups are typically older than infants with craniosynostosis who would be managed expectantly through established pathways. Please describe the cohort of patients who would be eligible for direct ICP assessment through your
--

	institution's clinical protocols, and how this might influence the generalizability or applicability of your results. 3. Describing the anticipated limitations of this study would increase its credibility.
--	---

VERSION 1 – AUTHOR RESPONSE

Suggestion, Question, or Comment from Reviewer 1: Dr. Hanna Zimmermann	Authors' Response	Manuscript section featuring changes
This is an interesting a highly relevant study protocol. However, I think some processes should be described more clearly. Also I think the authors could still optimize their analysis strategy.	We thank Dr. Hanna Zimmerman for this positive feedback, guidance and expertise. We have addressed all suggestions systematically in this table.	Please see below.
The definition of outcomes is not very clear. „The primary outcome measures are handheld OCT and 48-hour ICP assessments,...“ and later the authors list almost 20 different OCT parameters, However, „Primary outcome“ is supposed to be exactly 1 measure, so the authors might consider to define a specific OCT parameter or correlation as primary outcome.	Thank you – we have clarified as follows: “Our primary OCT outcome measure is retinal nerve fibre layer (RNFL) thickness; other OCT parameters are listed below.”	Methods: Outcome measures, pg. 5
The use of the \pm symbol is not very clear to me. E.g. „HH-OCT will be performed during initial and all follow-up appointments until 48-hour ICP monitoring + surgery + follow-up appointments where applicable.“ Please provide a very clear description of the baseline and follow up times, preferably as a chart. I think it is important that the time points are very standardized in order to allow for comparability.	Many thanks for this point. We have simplified matters as follows: “Handheld OCT will be performed during 48-hour ICP monitoring to fulfil our primary objective; wherever possible, handheld OCT will also be performed during clinic visits to fulfil our secondary objectives.” I.e. only the OCTs taken during 48-hour ICP monitoring are	Methods: Handheld optical coherence tomography image acquisition and analysis, pg. 6 para. 1

	included in our diagnostic accuracy testing.	
The authors describe they are using a 12x8 mm scanning field for OCT. This is quite large, is this supposed to cover the macula and ONH in one scan, or are both regions scanned with this protocol individually?	Yes – we have clarified as follows: “A 12×8-mm scanning window will be used in the acquisition protocol – this permits imaging of both the ONH and fovea in one scan.” This is the same protocol we have used in our feasibility study, which we have referenced.	Methods: Handheld optical coherence tomography image acquisition and analysis, pg. 6 para. 1
Regarding OCT image analysis, I understand that the device they are using delivers no parameters and all analysis is done manually in ImageJ. This should be described in more detail.	Thank you – we have substantially expanded upon this section to describe our OCT image analysis methodology in detail.	Methods: Handheld optical coherence tomography image acquisition and analysis, pg. 7
While I agree with the authors that results on OCT in children with IP are sparse, there is a plethora of studies investigating OCT in adult patients. I believe it makes sense to refer more to those. Especially, the 3D analysis of ONH volume has been of particular diagnostic value:	Many thanks for suggesting this - we have referenced these studies. We have also included our more recent systematic review (August 2021, BMJ Open), featuring 21 studies using OCT in children at risk of IH with craniosynostosis, IIH, space occupying lesion and other pathology, which also includes further discussion of adult studies.	Introduction: Optical coherence tomography, pg. 3
 - Wang J-K, Kardon RH, Kupersmith MJ, Garvin MK. Automated quantification of volumetric optic disc swelling in papilledema using spectral-domain optical coherence tomography. Invest Ophthalmol Vis Sci. 2012; 53: 4069–4075. - Albrecht P, Blasberg C, Ringelstein M, et al. Optical coherence tomography for the diagnosis and monitoring of idiopathic intracranial hypertension. J Neurol. 2017; 264: 1370–1380. - Kaufhold F, Kadas EM, Schmidt C, et al. Optic nerve 		

head quantification in idiopathic intracranial hypertension by spectral domain OCT. PLoS One. 2012; 7: e36965.		
While I understand that the HH-OCT device that is used does not provide volumetric analysis, the authors might consider to implement a method for that, in order to make maximum use of the data assessed.	Many thanks for this point. We have clarified as follows: “Full peripapillary volumetric analysis can be performed wherever possible, using a recently published protocol⁵¹” This protocol was developed by two of the senior authors from our group – Dr. Proudlock and Prof. Gottlob.	Methods: Handheld optical coherence tomography image acquisition and analysis, pg. 8
As visual fields demand a high degree of compliance and patient cooperation: Have the authors considered to use multifocal VEP as an objective measure of perimetry?	Thank you for this suggestion. We have consulted the opinion of our specialist visual electrophysiologist who advised that multifocal VEP requires very accurate fixation, hence he feels it is equally unlikely to be achieved in this patient population. We agree that the visual fields success rate will be low, hence we have stated “where possible” for this and all secondary outcome measures.	Addressed in Methods: Outcome measures, pg. 6
Acquisition of high-quality OCT data in the targeted patient cohort will be challenging because of the axial expansion of papilledema, and the limited compliance of children. How are the authors going to address this?	Thank you for this query. We have clarified as follows: In the Introduction, we have added the findings from our recent feasibility and repeatability study, including factors for success. In Methods, we have provided more details of our handheld OCT image acquisition methods, including the use of cartoons and toys as visual fixation devices as appropriate	Introduction: Handheld optical coherence tomography, pg. 4, para. 2 Methods: Handheld optical coherence tomography image acquisition and analysis, pg. 7, para. 1 and 2

	and an assistant where available. In Methods, we have described our technique to ensure the entire ONH is visualised even in cases of papilloedema, which was successful in our recent feasibility and repeatability study.	
How are the authors going to address the growing eyeball size in children?	Thank you for this important point, which we have clarified as follows: “Lateral distance measurements (defined for adults on the machine) shall be corrected to account for the smaller axial lengths in children using a conversion table according to age from the data presented by Maldonado et al.⁵⁰” Please note we also addressed this issue in our recent feasibility and repeatability study (TVST, July 2021).	Methods: Handheld optical coherence tomography image acquisition and analysis, pg. 7, para. 3
Statistical analysis: Can the authors comment how they are going to address intra-subject inter-eye correlations?	Thank you for this important query. We have clarified that the eye recorded (right/left) will be included in the regression model, to address inter-eye correlations.	Methods: Statistical analysis, pg. 9
General: Please make sure to introduce acronyms when they are first mentioned, e.g. „IIH“ in abstract	Thank you – we have revised accordingly.	Abstract, pg. 2 Introduction, pg. 3

Suggestion, Question, or Comment from Reviewer 2, Dr. Jordan Swanson	Authors' Response	Manuscript section featuring changes
The authors seek to assess the use of handheld optical coherence tomography (OCT) to detect elevated intracranial pressure (ICP) in at-risk children, by examining retinal parameters in children who are undergoing direct ICP measurement. The study is well-conceived, and builds on evolving evidence which the authors present articulately. It has the potential to improve both clinical pathways and treatment decision-making in unique clinical cases, by improving the fidelity of non-invasive methods to evaluate ICP. Several additional considerations may strengthen the manuscript.	We thank Dr. Jordan Swanson for this positive feedback, guidance and expertise. We have addressed all suggestions systematically in this table.	Please see below
Pediatric ophthalmology centers (including ours) report difficulty and inconsistency in obtaining OCT data from awake children under 3 years of age. Although the authors cite two references from their team (30,35) of “reliably” using OCT in infants, it is not clear from either of these references the “success rate” of success compared to attempts at OCT acquisition in infants. This information, and/or practical guidance for how this can be feasibly undertaken, would strengthen the methods section. Does the number of prospective subjects to include need to be increased to account for this?	Many thanks for raising this. To satisfy this query, fortunately we have just published our feasibility and repeatability study, the findings of which we have now included in this protocol manuscript: “More recently, we performed a feasibility and repeatability study⁴⁷ using handheld OCT in children with craniosynostosis (n=50, median age 51 months, age range 2–157 months). We found good feasibility, with 86% achieving at least unilateral OCT and 76% achieving bilateral OCTs of the optic nerve head (ONH). This was higher than the success rate in healthy children found by Patel et al³⁷ (70% achieving at least unilateral OCTs of the ONH). Factors boosting the likelihood of success in children with craniosynostosis included good	Introduction: Handheld optical coherence tomography, pg. 4 para. 2

	understanding and cooperation of the child and parent/guardian and availability of an assistant. We also performed repeatability analysis in 20 children and found good repeatability (intraclass correlation coefficient [ICC] range, 0.77–0.99; the majority exceeded 0.90).”	
Direct ICP assessment in patients with craniosynostosis at many craniofacial centers is obtained selectively for atypical patients and those with suspected recurrence of cranio-cerebral disproportion. Both of these groups are typically older than infants with craniosynostosis who would be managed expectantly through established pathways. Please describe the cohort of patients who would be eligible for direct ICP assessment through your institution’s clinical protocols, and how this might influence the generalizability or applicability of your results.	Many thanks for this suggestion. We have clarified as follows: “In Great Ormond Street Hospital for Children (GOSH), London, ICP bolt (Raumedic AG, Helmbrechts, Germany) monitoring is performed for 48 hours according to clinical discretion, particularly in children with clinical suspicion of IH where ophthalmological findings, including fundoscopic and electrodiagnostic findings, are equivocal. This represents an important patient cohort where a more sensitive non-invasive measure could improve clinical decision making and reduce the need for ICP bolt monitoring. However, the youngest in this group are typically toddlers and older, whereas those aged under 1 year with conditions such as craniosynostosis typically undergo surgical treatment expectantly rather than having ICP bolt monitoring.”	Introduction: Measuring intracranial pressure, pg. 3
Describing the anticipated limitations of this study would increase its credibility.	Many thanks for this advice. We have added the following limitations in the Strengths and Limitations section:	Strengths and limitations, pg. 2

	 • Fully automated handheld OCT image analysis is not currently available and manual segmentation takes longer, albeit our evidence-based methodology features semi-automated segmentation. • Although handheld OCT can provide in vivo imaging in infants without sedation, it could be especially challenging in this patient population, which could limit imaging success rates. • Gold standard intracranial pressure measurements form our reference standard, but these are only performed in select patients, thus limiting our overall recruitment. We note that the BMJ Open author guidelines for Protocol articles stipulates that limitations are included as bullet points in this section rather than a subsection within the main text. We shall write a more detailed Limitations section in the manuscript of our actual study, once complete.	
--	--	--

VERSION 2 – REVIEW

REVIEWER	Zimmermann, Hanna Charité Universitätsmedizin Berlin
REVIEW RETURNED	13-Oct-2021
GENERAL COMMENTS	The authors have addressed all my concerns. I have no more comments.